# Patterns of multimorbidity and their effects on adverse outcomes in rheumatoid arthritis: a study of 5658 UK Biobank participants

Ross McQueenie [1,2] Barbara I Nicholl,[1] Bhautesh D Jani,[1] Jordan Canning,[1] Sara Macdonald,[1] Colin McCowan,[3] Joanne Neary,[1] Susan Browne,[1] Frances S Mair,[1] Stefan Siebert [4]

► Prepublication history and additional materials for this paper is available online. To view these files, please visit the journal online (http://dx.doi.org/10.1136/bmjopen-2020-038829).

FSM and SS contributed equally.

[1]General Practice and Primary Care, Institute of Health and Wellbeing, MVLS, University of Glasgow, Glasgow, UK
[2]Public Health Scotland, NHS Scotland, Glasgow, UK
[3]School of Medicine, University of St. Andrews, Saint Andrews, UK
[4]Institute of Infection, Immunity and Inflammation, University of Glasgow, Glasgow, UK

**Correspondence to**
Prof Stefan Siebert;
Stefan.Siebert@glasgow.ac.uk

## ABSTRACT

**Objective** To investigate how the type and number of long-term conditions (LTCs) impact on all-cause mortality and major adverse cardiovascular events (MACE) in people with rheumatoid arthritis (RA).

**Design** Population-based longitudinal cohort study.

**Setting** UK Biobank.

**Participants** UK Biobank participants (n=502 533) aged between 37 and 73 years old.

**Primary outcome measures** Primary outcome measures were risk of all-cause mortality and MACE.

**Methods** We examined the relationship between LTC count and individual comorbid LTCs (n=42) on adverse clinical outcomes in participants with self-reported RA (n=5658). Risk of all-cause mortality and MACE were compared using Cox's proportional hazard models adjusted for lifestyle factors (smoking, alcohol intake, physical activity), demographic factors (sex, age, socioeconomic status) and rheumatoid factor.

**Results** 75.7% of participants with RA had multimorbidity and these individuals were at increased risk of all-cause mortality and MACE. RA and ≥4 LTCs showed a threefold increased risk of all-cause mortality (HR 3.30, 95% CI 2.61 to 4.16), and MACE (HR 3.45, 95% CI 2.66 to 4.49) compared with those without LTCs. Of the comorbid LTCs studied, osteoporosis was most strongly associated with adverse outcomes in participants with RA compared with those without RA or LTCs: twofold increased risk of all-cause mortality (HR 2.20, 95% CI 1.55 to 3.12) and threefold increased risk of MACE (HR 3.17, 95% CI 2.27 to 4.64). These findings remained in a subset (n=3683) with RA diagnosis validated from clinical records or medication reports.

**Conclusion** Those with RA and other LTCs, particularly comorbid osteoporosis, are at increased risk of adverse outcomes, although the role of corticosteroids could not be evaluated in this study. These results are clinically relevant for the monitoring and management of RA across the healthcare system, and future clinical guidelines for RA should acknowledge the importance of multimorbidity.

## Strengths and limitations of this study

► This is the first study to examine both comorbidity and multimorbidity in rheumatoid arthritis (RA) and the associations with mortality and major adverse cardiovascular events.

► We used data from 5658 participants in UK Biobank with RA, including detailed information on participant demographics, lifestyle factors and rheumatoid factor status to examine multimorbidity and comorbidity using 42 non-RA long-term condition.

► These results provide crucial new information which should be incorporated into clinical guidelines and used to influence management of peoples with RA.

► This study was limited by lack of information on RA disease severity which may play a role in both outcomes measured.

## INTRODUCTION

Rheumatoid arthritis (RA) is a debilitating, chronic autoimmune disease characterised by inflammation of the synovial joints. RA is associated with physical and socioeconomic issues, including increased pain levels, reduced physical functioning and early mortality.[1–4] Globally, while disability adjusted life years for RA have improved since 1990, age-standardised prevalence and incidence rates are increasing.[5]

Between 60% and 75% of those with RA are reported to have multimorbidity—two or more long-term conditions (LTCs)—with higher number of LTCs reported with increasing age and disease activity.[6–8] Common comorbidities include cardiovascular conditions[9] such as coronary artery disease[10] and cardiac failure,[11] as well as mental health conditions such as depression.[12] Cardiovascular disease accounts for the majority of the excess mortality observed in RA, with raised inflammatory markers and shared risk factors implicated.[13] However, the effects of comorbidities in RA have generally been studied in isolation and less is known regarding the risks

posed by multimorbidity when RA co-occurs with more than one other long-term physical or mental health LTC.

Through analysis of UK Biobank data, this paper aims to explore the effect of multimorbidity and a wide range of comorbid LTCs on all-cause mortality and major adverse cardiovascular events (MACE) in people with RA. Our objectives were to:

1. Compare the effect of LTC count on all-cause mortality in those with and without self-reported RA.
2. Compare the effect of LTC count on MACE in those with and without self-reported RA.
3. Evaluate the effect of individual comorbid LTCs on the risk of all-cause mortality and MACE in participants with self-reported RA.

## PATIENTS AND METHODS
### Study design and data collection
This study used data from UK Biobank, a longitudinal population-based cohort of 502 533 participants, aged 37–73 years in Great Britain.[14] UK Biobank baseline data were collected between 2006 and 2010 from recruitment centres in Scotland, England and Wales, and subsequently linked to mortality and hospitalisation outcomes from external routine data registries over a median follow-up period of 9 years. A subset of primary care data was available for 230 105 participants.

### Variables and outcome measures
UK Biobank collected information on a wide range of demographic, health-based lifestyle and self-reported LTC questions through self-administered touch screen questionnaire and nurse-led interview. These include age, sex, socioeconomic status (measured using Townsend score, a UK area-based measure of deprivation),[15] smoking status, frequency of alcohol intake, body mass index (BMI), level of physical activity and number of LTCs.

The age range of the study population was 37–73 years and was categorised into groups: 37–49, 50–59 and 60–73 years. Sex was a binary categorical variable. Smoking status was categorised into 'never' or 'current or previous'. Frequency of alcohol intake was categorised into four groups, 'Never or special occasions only', 'One to three times a month', 'One to four times a week' or 'Daily or almost daily'. BMI was categorised into four groups based on WHO BMI guidelines[16]: 'underweight <18.5', 'normal weight 18.5–24.9', 'overweight 25–29.9' and 'obese ≥30'. Level of physical activity was defined as 'none', 'low', 'medium', or 'high' using Metabolic Equivalent Task scores data based on International Physical Activity Questionnaire scoring protocol (available from https://sites.google.com/site/theipaq/scoring-protocol) which has shown moderate to good validity and reliability in adults in UK settings.[17 18]

Rheumatoid factor was ascertained, as part of a predefined biomarker panel, for all participants in UK Biobank, regardless of diagnosis, and categorised into positive and negative status, with rheumatoid factor <20 IU/mL considered negative, and values above this considered positive (by manufacturer specification, available at https://www.beckmancoulter.com/wsrportal/techdocs?docname=/cis/988646/%/RF_988646-%25%25_English.pdf). Participants whose rheumatoid factor was labelled as 'not reportable at assay (too low)' were considered to be rheumatoid factor negative. Similarly, those labelled 'not reportable at assay (too high)' were considered rheumatoid factor positive.

The list of 42 LTCs considered was based on previous work in UK Biobank,[19 20] the number of LTCs reported, apart from RA, were summed and then categorised as 0, 1, 2–3 and ≥4 LTCs. RA and all LTCs in UK Biobank are based on self-report using a questionnaire and nurse-led interview asking for existing diagnoses.

All-cause mortality was calculated using data linkage to national mortality registers. MACE were calculated using stroke and myocardial infarction hospitalisation event data from UK Biobank, and using International Classification of Diseases, 10th revision (ICD-10) mortality codes: 'I00-I78', 'G45', 'G451-G454', 'G456', 'G458', 'G459', and 'G460-G468'. The median follow-up time for both morality and MACE was 9 years; the length of follow-up for each participant varied as follow-up continued until an event occurred (death or MACE) or until the mortality the linkage was carried out.

A sensitivity analysis of self-report RA by participants was performed by examining four other indicators of RA: any primary care RA Read code, any secondary care RA hospitalisation code, self-reporting of any common RA drugs or any primary care prescription record of RA drugs (as shown in online supplemental table 1). Both prospective and retrospective data were used: primary care Read codes were available for a maximum period of January 1991 and December 2017, and primary care prescriptions were between January 1991 and December 2016; the time period for each participant varied, depending on records held. Participants were considered to have confirmed RA if they had a positive record for one or more of these indicators. This analysis was performed on a subset (74%) of participants who self-reported RA for whom primary care data in UK Biobank was available (n=4196/5658).

### Statistical methods
In line with previous UK Biobank studies, $\chi^2$ tests were used for both categorical data and ordinal data. Kruskal-Wallis tests were used for continuous data.[21] Similarly, we used $\chi^2$ testing to examine differences in proportion of individual LTCs between those with and without RA. Age-adjusted Cox's proportional hazards tests were used to examine the relationship between LTC count/type of LTCs with all-cause mortality and MACE as outcome variables in those with and without RA. The model was further adjusted for demographic, lifestyle and biological factors (sex, Townsend score, alcohol status, smoking status, BMI, physical activity and rheumatoid factor status) as described above. Among those with RA, cumulative hazards-based Kaplan-Meier plots were used

to display proportion of events (all-cause mortality or MACE) in participants with 0, 1, 2–3 and ≥4 comorbid LTCs. To measure the contribution of individual index LTCs towards all-cause mortality and MACE in those with and without RA, we created a categorical variable that assigned participants to one of four groups: those with neither RA nor the index condition (reference group), those with RA but not the index LTC (RA only), those with no RA with the index LTC (index LTC only), and those with both RA and the index LTC. This variable was used as an outcome measure in an age-adjusted Cox's proportional hazards model controlling for demographic factors, lifestyle factors and rheumatoid factor status. To calculate whether there was a multiplicative or synergistic effect between RA and each index LTC, we used an analysis of variance (ANOVA) to compare the p alues between two Cox's proportional hazards models: the first contained RA and the index LTC, and the second contained RA, the index LTC and a statistical interaction term between RA and the index LTC. A statistical interaction was considered significant when the ANOVA test has a p<0.01.

## RESULTS

5658 UK Biobank participants (1.1%) reported having RA. Lifestyle and demographic characteristics of participants with and without self-reported RA are shown in table 1. Participants with RA were significantly more likely to be older, women, have lower socioeconomic status, be current or previous smokers, have a lower frequency of alcohol intake, have a BMI ≥30, have lower levels of physical activity and have larger numbers of comorbid LTCs. $\chi^2$ testing showed participants with self-reported RA were significantly more likely to have rheumatoid factor positive status: 35.6% had rheumatoid factor levels of over 20 IU/mL—compared with 3.6% in those without RA.

### Prevalence of LTCs in people with RA

Proportions of number of LTCs in participants with and without RA are shown in table 1. Reporting multiple LTCs was more common in those with RA: 34.5% had 2–3 LTCs (27.1% in those without RA), and 11.1% had ≥4 LTCs (4.9% in those without RA). Overall, 75.7% of people with RA were noted to be multimorbid. The difference in comorbidity experienced by those with and without RA is shown in online supplemental table 2). Those with RA reported proportionately higher numbers of physical and mental health-based LTCs, namely: cardiovascular LTCs including hypertension, coronary heart disease and stroke or transient ischaemic attack; pulmonary LTCs including asthma, chronic obstructive pulmonary disease (COPD) and chronic bronchitis; digestive system LTCs including dyspepsia, irritable bowel syndrome and inflammatory bowel disease; musculoskeletal conditions including osteoporosis; and mental-health based LTCs including depression.

**Table 1** Demographic factors, lifestyle factors, number of long-term conditions and rheumatoid factor status in patients with and without rheumatoid arthritis (RA)

| | Participants with RA (n=5658) | Participants without RA (n=496 882) |
|---|---|---|
| Mean age (years (SD)); missing values=0 (0%) | 59.3 (7.1) | 56.5 (8.1) |
| Age (years); missing values=0 (0%) | | |
| 37–49 | 675 (11.9%) | 117 209 (23.6%) |
| 50–59 | 1800 (31.8%) | 165 359 (33.3%) |
| 60–73 | 3183 (56.3%) | 214 314 (43.1%) |
| Sex; missing values=0 (0%) | | |
| Female | 3952 (69.8%) | 269 452 (54.2%) |
| Male | 1706 (30.2%) | 227 430 (45.8%) |
| Townsend score; missing values=623 (0.12%) | | |
| 0–20 (least deprived) | 998 (17.7%) | 99 665 (20.1%) |
| 20–40 | 980 (17.4%) | 99 117 (20%) |
| 40–60 | 1087 (19.2%) | 99 311 (20%) |
| 60–80 | 1154 (20.4%) | 99 224 (20%) |
| 80–100 (most deprived) | 1429 (25.3%) | 98 952 (19.9%) |
| Smoking status; missing values=2950 (0.59 %) | | |
| Never | 2625 (46.8%) | 270 916 (54.8%) |
| Current or previous | 2983 (53.2%) | 223 066 (45.2%) |
| Frequency of alcohol intake; missing values=1502 (0.30 %) | | |
| Never or special occasions only | 1830 (32.4%) | 96 832 (19.5%) |
| One to three times a month | 690 (12.2%) | 55 170 (11.1%) |
| One to four times a week | 2315 (41%) | 242 428 (48.9%) |
| Daily or almost daily | 811 (14.4%) | 100 962 (20.4%) |
| BMI (kg/m$^2$); missing values=5820 (1.15%) | | |
| Underweight <18.5 | 50 (0.9%) | 2576 (0.5%) |
| Normal weight 18.5–24.9 | 1543 (27.9%) | 155 896 (31.7%) |
| Overweight 25–29.9 | 2194 (39.6%) | 212 032 (43.2%) |
| Obese ≥30 | 1750 (31.6%) | 120 679 (24.6%) |
| Physical activity; missing values=7156 (1.42 %) | | |
| None | 814 (14.8%) | 32 035 (6.5%) |
| Low | 409 (7.4%) | 18 531 (3.8%) |
| Medium | 4111 (74.5%) | 389 412 (79.5%) |
| High | 182 (3.3%) | 49 890 (10.2%) |
| Number of long-term conditions; missing values=1845 (0.36 %) | | |
| 0 | 1369 (24.3%) | 173 846 (35.1%) |

Continued

**Table 1** Continued

|  | Participants with RA (n=5658) | Participants without RA (n=496 882) |
|---|---|---|
| 1 | 1690 (30.0%) | 162 657 (32.9%) |
| 2–3 | 1943 (34.5%) | 134 403 (27.1%) |
| ≥4 | 623 (11.1%) | 24 157 (4.9%) |
| Rheumatoid factor (IU/ml); missing values=33 066 (6.6 %) | | |
| <20 | 3396 (64.4%) | 447 472 (96.4%) |
| >20 | 1879 (35.6%) | 16 720 (3.6%) |

Unless otherwise indicated, all results are shown as number (%). Unless indicated, p<0.01. $\chi^2$ test was used for categorical variables, Kruskal-Wallis test was used for continuous variables.

### All-cause mortality and LTCs in people with RA

We examined the outcomes associated with different LTC counts in participants with RA using a Kaplan-Meier plot (see online supplemental figure 1). There was an increased proportion of all-cause mortality in participants with RA concurrent with increasing multimorbidity counts: 4.2% (n=58) in those with no additional LTCs, 5.3% (n=91) in those with 1 additional LTC, 9.9% (n=194) in those with 2–3 additional LTCs and 14.4% (n=90) in those with ≥4 additional LTCs during the follow-up period (median 9 years).

To quantify the effect of LTC count on all-cause mortality, we performed a Cox's proportional hazards test controlling for lifestyle factors, demographic factors and rheumatoid factor in participants with and without self-reported RA using a stepwise model adjustment (table 2).

Participants with RA and no additional LTCs had a significant increase in all-cause mortality when using an age-adjusted Cox's proportional hazards model fully adjusting for additional lifestyle and demographic factors (HR 1.59, 95% CIs 1.21 to 2.08) compared with those without RA or any LTCs. While controlling additionally for rheumatoid factor status appeared to show some attenuation of all-cause mortality risk, a statistically significant risk for this group remained (HR 1.39, 95% CI 1.05 to 1.84) when compared with those without RA or any LTCs. When examining additional comorbid LTCs alongside RA, there appeared to be a dose-based response all-cause mortality risk, with a 44% increased risk of all-cause mortality in those with RA and one other LTC (HR 1.44, 95% CI 1.14 to 1.81), an approximately two-and-a-half-fold increased risk for RA with 2–3 other LTCs (HR 2.48, 95% CI 2.12 to 2.90) and an over threefold increased risk associated for RA with ≥4 other LTCs (HR 3.30, 95% CI 2.61 to 4.16) compared with those without RA or any LTCs in the fully adjusted models, which included rheumatoid factor. A dose-based response was also observed in the non-RA population: those with 1 LTC had a 39% increased risk of death (HR 1.39, 95% CI 1.33 to 1.46), and those with ≥4 were at a two-and-a-half-fold increased risk (HR 2.69 95% CI 2.54 to 2.85) compared with participants without RA or any LTCs.

### MACE and LTCs in people with RA

We next investigated the effect of LTC count on MACE in participants with RA using a Kaplan-Meier plot (see online supplemental figure 2). For RA and no additional LTCs, 3.3% (n=46) of participants had a recorded MACE

**Table 2** Relationship between long-term conditions and all-cause mortality in participants with and without self-reported rheumatoid arthritis (RA) using age-adjusted multivariate Cox's proportional hazards regression analysis

**Risk of all-cause mortality**

| Comorbidity status (reference: no RA and no other long-term conditions) | | Adjusted for sex and Townsend score HR (95% CI) | Adjusted for sex, Townsend score, alcohol status and smoking status HR (95% CI) | Adjusted for sex, Townsend score, alcohol status, smoking status, BMI and physical activity HR (95% CI) | Adjusted for sex, Townsend score, alcohol status, smoking status, BMI, physical activity and rheumatoid factor status HR (95% CI) | Number of deaths (%) |
|---|---|---|---|---|---|---|
| No other long-term conditions | RA | 1.84 (1.42 to 2.38) | 1.72 (1.32 to 2.2) | 1.59 (1.21 to 2.08) | 1.39 (1.05 to 1.84) | 58 (4.2%) |
| 1 other long-term condition | No RA | 1.45 (1.39 to 1.51) | 1.42 (1.36 to 1.48) | 1.40 (1.34 to 1.47) | 1.39 (1.33 to 1.46) | 5785 (3.6%) |
|  | RA | 2.01 (1.64 to 2.48) | 1.88 (1.53 to 2.32) | 1.72 (1.38 to 2.14) | 1.44 (1.14 to 1.81) | 91 (5.4%) |
| 2–3 other long-term conditions | No RA | 2.03 (1.95 to 2.11) | 1.92 (1.84 to 2.00) | 1.84 (1.77 to 1.92) | 1.83 (1.75 to 1.91) | 7914 (5.9%) |
|  | RA | 3.32 (2.87 to 3.84) | 2.99 (2.59 to 3.46) | 2.79 (2.40 to 3.24) | 2.48 (2.12 to 2.90) | 194 (10.0%) |
| ≥4 other long-term conditions | No RA | 3.39 (3.22 to 3.57) | 3.04 (2.88 to 3.20) | 2.71 (2.56 to 2.86) | 2.69 (2.54 to 2.85) | 2605 (10.8%) |
|  | RA | 4.68 (3.80 to 5.78) | 3.95 (3.19 to 4.89) | 3.52 (2.81 to 4.40) | 3.30 (2.61 to 4.16) | 90 (14.4%) |

Unless otherwise shown, Cox's proportional hazards p<0.01.
BMI, body mass index.

**Table 3** Relationship between long-term conditions and major adverse cardiovascular events in participants with and without self-reported rheumatoid arthritis (RA) using age-adjusted multivariate Cox's proportional hazards regression analysis

**Risk of MACE**

| Comorbidity status (reference: no RA and no other long-term conditions) | | Adjusted for sex and Townsend score HR (95% CI) | Adjusted for sex, Townsend score, alcohol status and smoking status HR (95% CI) | Adjusted for sex, Townsend score, alcohol status, smoking status, BMI and physical activity HR (95% CI) | Adjusted for sex, Townsend score, alcohol status, smoking status, BMI, physical activity and rheumatoid factor status HR (95% CI) | Number of MACE (%) |
|---|---|---|---|---|---|---|
| No other long-term conditions | RA | 1.79 (1.33 to 2.39) | 1.69 (1.26 to 2.27) | 1.64 (1.21 to 2.20) | 1.63 (1.21 to 2.21) | 46 (3.4%) |
| 1 other long-term condition | No RA | 1.30 (1.24 to 1.36) | 1.28 (1.22 to 1.34) | 1.26 (1.20 to 1.320 | 1.24 (1.19 to 1.31) | 4512 (2.8%) |
| | RA | 2.08 (1.66 to 2.61) | 1.91 (1.52 to 2.41) | 1.87 (1.48 to 2.35) | 1.68 (1.31 to 2.15) | 78 (4.6%) |
| 2–3 other long-term conditions | No RA | 1.86 (1.78 to 1.94) | 1.78 (1.70 to 1.86) | 1.67 (1.60 to 1.75) | 1.66 (1.59 to 1.74) | 6208 (4.6%) |
| | RA | 2.72 (2.28 to 3.24) | 2.49 (2.09 to 2.98) | 2.19 (1.82 to 2.64) | 2.09 (1.73 to 2.54) | 131 (6.7%) |
| ≥4 other long-term conditions | No RA | 3.04 (2.87 to 3.22) | 2.76 (2.60 to 2.93) | 2.40 (2.26 to 2.56) | 2.37 (2.23 to 2.53) | 1980 (8.2%) |
| | RA | 4.79 (3.79 to 6.04) | 4.07 (3.21 to 5.16) | 3.52 (2.73 to 4.52) | 3.39 (2.61 to 4.40) | 73 (11.7%) |

Unless otherwise shown, Cox's proportional hazards p<0.01.
BMI, body mass index; MACE, major adverse cardiovascular events.

event, compared with 4.6% of participants with RA and one additional LTC (n=78), 6.7% those with RA and 2–3 additional LTCS (n=131), and almost four times as many proportionately in participants with RA and ≥4 LTCs (11.7%, n=73 events) over the follow-up period.

Table 3 shows the risk of MACE for participants with and without RA using age-adjusted multivariate Cox's proportional hazards regression models. There was a 63% increased hazard of MACE for participants with RA and no other LTCs compared with participants without RA or any LTCs (HR 1.63, 95% CI 1.21 to 2.21) in a fully adjusted model including demographic factors, lifestyle factors and rheumatoid factor status. This remained significant for people with RA with increasing LTCs count, with an 86% increased risk of MACE in participants with one other co-occurring LTC (HR 1.86, 95% CI 1.31 to 2.15), an over twofold increase in those with 2–3 co-occurring LTCs (HR 2.09, 95% CI 1.73 to 2.54) and an almost three-and-a-half-fold increase in MACE for those with ≥4 LTCs (HR 3.39, 95% CI 2.61 to 4.40), compared with those without RA or any LTCs. This relationship was similar but to a lesser degree for participants without RA, with those with 1 LTC at 24% increased risk (HR 1.24, 95% CI 1.19 to 1.31), those with 2–3 LTCs at a 66% increased risk (HR 1.66, 95% CI 1.59 to 1.74) and those with ≥4 LTCs at over two times risk (HR 2.37 95% CI 2.23 to 2.53) of MACE compared with those without LTCs.

A similar pattern was observed for the relationship between LTC count and mortality/MACE for the group without RA (see online supplemental figure 3 and 4).

## Contribution of individual LTCs to all-cause mortality and MACE in people with RA

Using an age-adjusted Cox's proportional hazards model, adjusting for demographic factors, lifestyle factors and rheumatoid factor status, we investigated the role individual LTCs play in risk of all-cause mortality and MACE, using participants with no RA and no index condition as the reference group (tables 4 and 5).

The presence of cardiovascular-based LTCs appeared to be a risk factor in those with RA for both all-cause mortality and MACE. Compared with those with no RA and no hypertension, RA with hypertension showing an over one-and-a-half-fold increased risk of all-cause mortality (HR 1.59, 95% CI 1.37 to 1.86) and an approximately twofold increased risk of MACE (HR 2.07, 95% CI 1.64 to 2.33).

Similarly, heart disease was associated with an over twofold increase for both all-cause mortality (HR 2.07, 95% CI 1.63 to 2.63) and MACE (HR 2.28 95% CI 1.76 to 2.98) in those with RA compared with those with no RA and no heart disease. However, there was no evidence of interaction between RA and either cardiovascular condition. While thyroid disorders showed no significant increased risk of all-cause mortality, they displayed an over twofold increased risk of MACE (HR 2.10, 95% CI 1.50 to 2.93) in those with RA compared with those without RA or thyroid disease but again there was no significant interaction between RA and thyroid disease and MACE event.

The co-occurrence of osteoporosis in participants with RA appeared to strongly influence both mortality and MACE; more than doubling all-cause mortality (HR 2.20,

**Table 4** Risk of all-cause mortality for individual index conditions in patients with rheumatoid arthritis (RA) and no index condition, RA with index condition, RA with no index condition and RA and index condition

**Risk of all-cause mortality**

| | No RA, no index condition | No RA, with index condition HR (95% CI) | RA, no index condition HR (95% CI) | RA and index condition HR (95% CI) |
|---|---|---|---|---|
| **Index condition** | | | | |
| Hypertension | 1 | 1.24 (1.21 to 1.28) | 1.29 (1.11 to 1.48) | 1.59 (1.37 to 1.86) |
| Coronary heart disease | 1 | 1.57 (1.50 to 1.65) | 1.26 (1.12 to 1.42) | 2.07 (1.63 to 2.63) |
| Diabetes | 1 | 1.68 (1.60 to 1.75) | 1.33 (1.18 to 1.48) | 1.83 (1.37 to 2.44) |
| Asthma | 1 | 1.10 (1.05 to 1.15) | 1.27 (1.13 to 1.42) | 1.56 (1.22 to 2.00) |
| Dyspepsia | 1 | 1.01 (0.97 to 1.06)* | 1.27 (1.14 to 1.43) | 1.45 (1.10 to 1.90) |
| Cancer | 1 | 2.50 (2.41 to 2.59) | 1.35 (1.20 to 1.52) | 3.04 (2.39 to 3.86) |
| Depression | 1 | 1.27 (1.20 to 1.35) | 1.29 (1.15 to 1.44) | 1.71 (1.21 to 2.42) |
| Thyroid disorder | 1 | 1.05 (0.98 to 1.12)* | 1.32 (1.18 to 1.47) | 1.14 (0.80 to 1.62)* |
| COPD | 1 | 2.11 (1.98 to 2.49) | 1.26 (1.13 to 1.42) | 2.68 (2.00 to 3.58) |
| Epilepsy | 1 | 1.81 (1.42 to 1.82) | 1.29 (1.15 to 1.43) | 2.86 (1.43 to 5.73) |
| Migraine | 1 | 0.85 (0.76 to 0.94) | 1.29 (1.16 to 1.44) | 1.09 (0.55 to 2.19)* |
| Psoriasis/eczema | 1 | 1.05 (0.98 to 1.14)* | 1.27 (1.14 to 1.42) | 1.88 (1.20 to 2.95) |
| Prostate disease | 1 | 0.83 (0.76 to 0.90) | 1.30 (1.17 to 1.45) | 0.90 (0.43 to 1.90)* |
| Osteoporosis | 1 | 1.26 (1.14 to 1.39) | 1.25 (1.12 to 1.40) | 2.20 (1.55 to 3.12) |
| Atrial fibrillation | 1 | 1.40 (1.45 to 1.57) | 1.30 (1.17 to 1.45) | 1.32 (0.50 to 3.52)* |
| Anxiety | 1 | 1.22 (1.10 to 1.35) | 1.30 (1.16 to 1.44) | 1.48 (0.67 to 3.30)* |
| Inflammatory bowel disease | 1 | 1.37 (1.20 to 1.57) | 1.30 (1.17 to 1.44) | 1.30 (0.54 to 3.11)* |
| Heart failure | 1 | 2.69 (2.22 to 3.25) | 1.29 (1.16 to 1.43) | 5.14 (2.14 to 12.38) |

Age-adjusted Cox's proportional hazards models were adjusted for sex, Townsend score, smoking status, alcohol intake frequency, body mass index, physical activity level and rheumatoid factor status. Cox's proportional hazards p<0·01, except for those labelled with *indicating p>0.01. Index conditions labelled ** have interaction term p<0.01.
COPD, Chronic Obstructive Pulmonary Disease.

95% CI 1.55 to 3.12), and resulting in an over three times higher risk of MACE (HR 3.17, 95% CI 2.17 to 4.64) compared with those without RA or osteoporosis. This increased risk in those with both RA and osteoporosis was greater than in those with RA but no osteoporosis or those with osteoporosis but no RA. Interaction terms for RA and osteoporosis showed no significant interaction with all-cause mortality (p=0.10), suggesting an additive effect only, but displayed a significant interaction with MACE (p<0.01), suggesting a multiplicative or synergistic effect in the association with MACE.

### Sensitivity analysis of RA self-report
To investigate sensitivity of self-report by participants with RA, we examined the proportion of people with any primary care RA Read code, any secondary care RA hospitalisation code, self-reporting of any common RA drugs and any primary care prescription record of RA drugs (see online supplemental table 1) for participants who had self-reported RA and had available primary care data available in UK Biobank (n=4196). Medications used here were previously reported by Siebert *et al*.[21] Using this method, we were able to identify RA medications,

hospitalisations or primary care Read code in 3683 (87.8%) participants (see online supplemental table 3). Analysis performed in this study was repeated in these participants and showed the same relationships as those reported above in n=5658 with self-report RA, with only small changes in HR observed (see online supplemental tables 4–8).

### DISCUSSION
Within UK Biobank, multiple LTCs was common in participants with RA, with approximately 75.7% reporting multimorbidity and 45% of participants reporting two or more additional LTCs alongside RA. In our fully adjusted modes, increasing LTC count was associated with increased mortality and MACE in people with RA. When examining individual LTCs, we observed hypertension, heart disease, osteoporosis and thyroid disorders to increase risk of adverse outcomes. Of these, osteoporosis was associated with one of the largest increases in both adverse outcomes measured: participants with both RA and osteoporosis were at over three times the risk of all-cause mortality and

**Table 5** Risk of major adverse cardiovascular events (MACE) for individual index conditions in patients with rheumatoid arthritis (RA) and no index condition, RA with index condition, RA with no index condition and RA and index condition

**Risk of MACE**

| | No RA, no index condition | No RA, with index condition HR (95% CI) | RA, no index condition HR (95% CI) | RA and index condition HR (95% CI) |
|---|---|---|---|---|
| Index condition | | | | |
| Hypertension | 1 | 1.50 (1.44 to 1.55) | 1.48 (1.25 to 1.75) | 1.97 (1.66 to 2.33) |
| Coronary heart disease | 1 | 1.89 (1.80 to 1.98) | 1.43 (1.45 to 1.63) | 2.28 (1.76 to 2.98) |
| Diabetes | 1 | 1.67 (1.58 to 1.75) | 1.49 (1.31 to 1.69) | 1.69 (1.19 to 2.39) |
| Asthma | 1 | 1.12 (1.06 to 1.18) | 1.43 (1.25 to 1.63) | 1.47 (1.09 to 1.98) |
| Dyspepsia | 1 | 1.14 (1.08 to 1.20) | 1.39 (1.22 to 1.58) | 1.85 (1.30 to 2.34) |
| Cancer | 1 | 1.11 (1.04 to 1.17) | 1.43 (1.26 to 1.62) | 1.44 (0.98 to 2.11)† |
| Depression | 1 | 1.25 (1.17 to 1.34) | 1.39 (1.22 to 1.58) | 2.06 (1.41 to 3.00) |
| Thyroid disorder | 1 | 1.14 (1.03 to 1.23) | 1.37 (1.20 to 1.55) | 2.10 (1.50 to 2.93) |
| COPD | 1 | 1.49 (1.37 to 1.62) | 1.40 (1.24 to 1.59) | 1.97 (1.33 to 2.92) |
| Epilepsy | 1 | 1.50 (1.30 to 1.73) | 1.41 (1.21 to 1.60) | 2.21 (0.83 to 5.88)† |
| Migraine | 1 | 0.99 (0.89 to 1.12)† | 1.40 (1.23 to 1.58) | 2.08 (1.12 to 3.87) |
| Psoriasis/eczema | 1 | 1.05 (0.96 to 1.14)† | 1.42 (1.26 to 1.61) | 1.23 (0.64 to 2.37)† |
| Prostate disease | 1 | 0.92 (0.83 to 1.00)† | 1.41 (1.25 to 1.60) | 1.27 (0.64 to 2.54)† |
| Osteoporosis* | 1 | 1.34 (1.18 to 1.53) | 1.25 (1.10 to 1.41) | 3.17 (2.17 to 4.64) |
| Atrial fibrillation | 1 | 1.41 (1.25 to 1.60) | 1.72 (1.53 to 1.93) | 2.67 (1.99 to 5.95) |
| Anxiety | 1 | 1.28 (1.14 to 1.43) | 1.40 (1.24 to 1.59) | 2.73 (1.30 to 5.72) |
| Inflammatory bowel disease | 1 | 1.09 (0.92 to 1.29)† | 1.42 (1.26 to 1.60) | 1.11 (0.36 to 3.44)† |
| Heart failure | 1 | 2.64 (2.15 to 3.24) | 1.41 (1.25 to 1.59) | 3.45 (1.11 to 10.70)† |

Age-adjusted Cox's proportional hazards models were adjusted for sex, Townsend score, smoking status, alcohol intake frequency, body mass index, physical activity level and rheumatoid factor status. Cox's proportional hazards p<0.01, except for those labelled with †indicating p>0.01. Index conditions labelled *have interaction term p<0.01.
COPD, Chronic Obstructive Pulmonary Disease.

two times the risk of compared with those with neither LTC. The negative effect of having both RA and osteoporosis was particularly evident in MACE outcomes, for which there was a significant interaction between RA and osteoporosis, suggesting a multiplicative or synergistic effect on MACE of having both these conditions together. The presence of hypertension or heart disease alongside RA increased the risk of mortality and MACE, in keeping with previous literature,[22 23] but there was no evidence of a synergistic effect.

To the best of our knowledge, this paper is the first to compare LTC count and type of comorbid LTCs and their association with all-cause mortality and MACE in men and women with RA after adjusting for a wide range of sociodemographic and lifestyle variables along with rheumatoid factor status. In our study, increasing LTC count resulted in adverse outcomes in participants with RA, with an increased rate of all-cause mortality and MACE.

We have shown that multimorbidity is common in participants with RA, with around 75% of participants with RA reporting one or more additional LTCs. This is in agreement with reported comorbidity rates of between 60% and 75% in those with RA,[6–8] although these studies typically examined a smaller number of LTCs than in this study. We have shown participants with RA and 2–3 other LTCs were at over twice the risk of all-cause mortality, while those with ≥4 more were over three times the risk compared with participants with no LTCs. This data provide evidence for the first time the increased risk of all-cause mortality in men and women with RA and multimorbidity. While previous work has highlighted an increased risk of mortality in patients with RA,[24 25] or specific comorbidities alongside RA—for example, in COPD[26] and depression[27]—these studies did not examine the effect of LTC count. One matched cohort study used a multimorbidity weighted index to study the effect of multimorbidity on mortality, but only examined effects in women.[28] Another examined LTCs using the Charlson Comorbidity Index[29]; however, this measure uses only 19 LTCs and the study examined only all-cause mortality outcomes. Our study is the first study of its type to link multimorbidity in RA with MACE outcomes. Existing research has highlighted that RA increases the risk of cardiovascular events, and that individual LTCs such as diabetes and hypertension are risk factors[30]; however, to date, no study has shown an association between

multimorbidity and MACE outcomes in people with RA. Collectively, the results presented here report for the first time the magnitude of adverse outcomes associated with multimorbidity in those with RA.

In keeping with previous studies,[8 31] we have shown that osteoporosis prevalence is increased in those with RA compared with those without RA. The results presented in this paper, however, are the first to link osteoporosis in those with RA to increased risk of adverse outcomes and the first to show significant interaction between both conditions and MACE outcomes. The reasons for this association are not clear and cannot be extrapolated from the available data, which does not include factors such as disease severity or duration. One possibility may be that corticosteroids and RA disease activity play a role: corticosteroids are associated with increased prevalence of osteoporosis[32]; people with RA with higher levels of disease activity are more likely to receive corticosteroids; both corticosteroid use and increased RA disease activity are reported to be associated with worse outcomes in mortality and MACE.[33 34]

Our study therefore has several strong clinical implications. Current National Institute for Health and Care Excellence (NICE) guidelines for RA suggest annual checks for the development of hypertension, ischaemic heart disease, osteoporosis and depression in RA,[35] but do not highlight the increased risk of the co-occurrence of these LTCs with RA nor the risk posed by multimorbidity in general. In addition, we have shown a greatly increased risk of adverse outcomes in people with osteoporosis and RA that merits further investigation.

Our study has several key strengths: UK Biobank is a large population-based study with several thousand participants reporting RA; the study setting encompasses three countries within the UK (Scotland, England and Wales); it includes details of participant demographic and lifestyle factors as well as rheumatoid factor levels, which allowed us to adjust for variables, which have not been explored in previous studies.

Our study is limited by self-reporting of RA and LTCs by these participants; however, recent studies have shown that self-report is a reliable method for reporting RA.[36] In this study, we additionally used four RA indicators (any primary care RA Read code, any secondary care RA hospitalisation code, self-reporting of any common RA drugs and any primary care prescription record of RA drugs) to validate self-reported RA. Using this validation approach, we found a positive verification rate (participants self-reporting RA with further RA indicators) of 87.8% (n=3683). Re-analysis of the subset of participants with RA (see online supplemental tables 4–8) who had a validated RA report showed only small changes to Cox's proportional hazards models, and observed effects were in agreement with the population who self-reported RA. This provides confidence in our findings that we are examining a true RA population. Rheumatoid factor positive status in those self-reporting RA (35.6%) was lower than expected, however still a significantly higher proportion

than in the UK Biobank population who did not report RA (3.6%). Analysis of rheumatoid factor in those who had a validated RA report showed an increased proportion of positive rheumatoid factor (47.6%), but this level remained below previously reported proportions in RA populations. We were unable to determine the severity or duration of RA in participants, or their previous medications. Participants in UK Biobank are known to be less deprived than the wider UK population,[37] suggesting that the level of multimorbidity reported here, and resulting associations are likely to be conservative in nature. Future work will examine potential clusters of LTCs that are associated with poor health-related outcomes in people with RA to try to inform clinical management of patients with RA and multiple LTCs.

## CONCLUSIONS

Multimorbidity is common in people with RA and is associated with increased risk of all-cause mortality and MACE. Certain comorbidities such as osteoporosis merit specific attention, in view of their association with adverse outcomes; it will be important to test whether this association is replicated in other datasets and if so, to explore the underpinning mechanisms. As multimorbidity has been shown here to influence outcomes for those with RA, forthcoming work will examine which clusters of LTCs most strongly drive this increased risk of poor outcomes. Future clinical guidelines for RA should acknowledge the importance of multimorbidity when considering management planning and patient outcomes.

**Acknowledgements** This research has been conducted using the UK Biobank Resource, approved project number 14151; the authors are like to thank the participants and those managing the data. They would like to acknowledge Versus Arthritis for funding this project (grant number 29170).

**Contributors** This study was conceived by BIN, FSM, SS, BDJ and CM. The analysis was conducted by RM, BIN and BDJ. All authors contributed to design, interpretation and discussion of all analysis. RM wrote this manuscript. All authors edited, reviewed and commented on all versions of this manuscript. All authors read the manuscript draft and approved the final submission.

**Funding** This study was funded by Versus Arthritis (grant number 21970).

**Competing interests** None declared.

**Patient and public involvement statement** The study was supported by a patient advisory group which provided input to the programme of research. This patient advisory group met on a regular basis for the duration of the study. Patients partnered with us and helped design research questions.

**Patient consent for publication** Not required.

**Ethics approval** All participants gave informed consent for data provision and linkage. UK Biobank has full ethical approval from the NHS National Research Ethics Service (16/NW/0274).

**Provenance and peer review** Not commissioned; externally peer reviewed.

**Data availability statement** Data may be obtained from a third party and are not publicly available. The data used in this study are available via a direct application to UK Biobank.

**ORCID iDs**
Ross McQueenie http://orcid.org/0000-0002-0070-6607
Stefan Siebert http://orcid.org/0000-0002-1802-7311

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
