## [Reviewer comments · BMJ Open]

ARTICLE DETAILS

TITLE (PROVISIONAL)	Patterns of multimorbidity and their effects on adverse outcomes in rheumatoid arthritis: a study of 5658 UK Biobank participants
AUTHORS	McQueenie, Ross; Nicholl, Barbara; Jani, Bhatesh; Canning, Jordan; MacDonald, Sara; McCowan, Colin; Neary, Joanne; Browne, Susan; Mair, Frances; Siebert, Stefan

VERSION 1 – REVIEW

REVIEWER	Wenhui Zhang Emory University, USA
REVIEW RETURNED	13-May-2020

GENERAL COMMENTS	Please see the comments in the file attached. The reviewer provided a marked copy with additional comments. Please contact the publisher for full details.
---

REVIEWER	Claire DAIEN Montpellier University and Hospital of Montpellier
REVIEW RETURNED	15-May-2020

GENERAL COMMENTS	The manuscript entitled "Patterns of multimorbidity and their effects on adverse outcomes in rheumatoid arthritis: a study of 5658 UK Biobank participants" describes very well the relationship between long-term conditions and the occurrence of MACES or death. It is very well written and very clear. I believe it would be very interesting to further explore and comment the interaction between RA on those LTCs on the outcomes. Is there any synergistic effect with RA or is it only additive? Please present the results so that this question is also addressed; e.g. in tables or in suppl fig, add also non -RA so we see if the curves RA vs non-RA get closer or not further in the different conditions (0, 1-2, 3 >4 LTCs). The main concern is the absence of data on glucocorticoids that may indeed be the most probable link between osteoporosis and MACES. This should be said in conclusion of abstract. Minor comment: please clarify mean follow-up of patients and add on figures the n of patients in each group at the different timepoints. Not sure * is used properly in table 4 and 5.
--

VERSION 1 – AUTHOR RESPONSE

Reviewer 1 comments (taken from PDF comments)

P4 In 32: Abstract: The RA abbreviation should be spelled out when first mentioned.
Thank you for highlighting this omission which we have added.

P4 In 34: Abstract: I thought this was a longitudinal study
You are correct. We apologise for this transcription error which we have corrected.

P7 In 97: Common comorbidity should also include obesity, diabetes
While these conditions may be associated with RA, they are not considered to be common comorbidities in RA (in contrast to psoriasis and psoriatic arthritis) so we have not added this.

P7 In 111: I would judge the reliability of self-reported arthritis
This was also a concern for us and, as you noted, was something we addressed with a sensitivity analysis (lines 296-305)

P8 In115: The data came from 2006-2010 and thus seems out-of-date
The UK Biobank baseline data was collected during 2006 to 2010, when participants attended an assessment centre to complete their assessment, this data was subsequently linked to external mortality and hospitalisations registries for a mean of 9 years of follow-up across the study population. In this study we have looked at associations between RA and LTCs reported at baseline and their mortality/MACE events in this follow-up period. We do not consider this data to be out of date; the earlier baseline has given us a longer period of follow-up to assess the relationships between RA, LTCs and health outcomes. We have added text to the manuscript to clarify that the 2006-2010 dates relate to UK Biobank baseline (line 117 and lines 119-120).

P8 In 119: Have you considered using any genetic data?
Thank you for this suggestion. We agree that UK Biobank includes valuable genetic data, however, this is not relevant to the question and issues being addressed in this manuscript, so is something we will consider in future studies.

P8 In 125: Why was age categorized this way?
UK Biobank intended to include participants aged 40-70 at baseline, however, there was a small number of participants who were included who were aged 37-39 and 71-73. Age bands of 40-49 years, 50-59 years, and 60-70 years did not quite capture the full cohort, hence extending the first to 37-49 years and the last 60-73 years, to allow for these outliers. The manuscript (line 129-130) has been updated for clarity: "The age range of the study population was 37-73 years and was categorised into groups: 37-49, 50-59 and 60-73 years."

P8 In 134: Remember to report the scales' reliability and validity.
The IPAQ reliability and validity consists of a number of measures that vary between and within countries, so we have added the following text in line 138-139: "which has shown moderate to good validity and reliability in adults in UK settings" and added the following references: Cleland et al BMC Methodology 2018;18:176 PubMed and Craig et al Med Sci Sports Exercise 2003;35:1381-95.

P9 In 138: Please expand and explain how rheumatoid factor is related to RA diagnosis
As outlined in the methodology and results sections of the manuscript, RA diagnosis in this study was self-reported (and validated using hospital and primary care data) and not related or dependent on rheumatoid factor (RF). RF was measured for all participants (regardless of diagnosis) in UK Biobank

as part of the biomarker panel. We have clarified this by amending line 141-142 as follows:
“Rheumatoid factor was ascertained, as part of a predefined biomarker panel, for all participants in UK Biobank, regardless of diagnosis, and categorised into...”

P9 In 144: Why were LTCs categorized in this way instead of using other numbers such as 5? Self-reported LTCs were summed and categorised as: 0, 1, 2-3 and ≥ 4 LTCs. This choice of categories was to reflect a range of LTC experience; with ≥ 4 LTCs suggesting a high level of multimorbidity. We did not categorise further than ≥ 4 LTCs as the number of participants with individual counts of 5, 6, 7... LTCs would not have allowed us to carry out detailed analysis.

P10 In 178-181: This was unclear

The description of the statistical interaction analysis has been updated for clarity (lines 190-195 of the revised manuscript).

P15 In 286: Nice catch!

Thank you.

P16 In 309: Make sure the citations are in correct formats

Thank you for highlighting this editorial issue – we have amended the formatting for all the citations to fit the journal style (we have not used track changes for this as it involves many changes that would make assessment of the manuscript too difficult).

P16 In 313: Have the authors checked gender differences?

Unfortunately we were not able to do stratified analysis by gender due to the number of events and the fact that 69% of the participants with RA are female. Please note that the current results are adjusted for gender.

P16 In 321: What were your findings on COPD and diabetes here?

As shown in tables 4 and 5, risk associated with presence of comorbid COPD and diabetes (and several other conditions) is additive but not multiplicative in nature.

P17 In 341: Were other disabling conditions explored? For example, visual impairment? Also, it seemed that obesity has not been controlled?

Other disabling conditions, such as visual impairment, were not in the predefined list of 42 LTCs and were not explored due to limited data in UK Biobank.

BMI was controlled for in the models, this omission was an oversight in our analysis section and this has now been updated on lines 179-180, instead of saying that we adjusted for demographic and lifestyle factors, we have added the following: “The model was further adjusted for demographic, lifestyle and biological factors (sex, Townsend score, alcohol status, smoking status, BMI, physical activity and rheumatoid factor status) as described above.”

P18 In 360-367: Try to clean and put limitations in one paragraph and also point out future directions.

The limitations section has been edited to shorten the length and to present it in one paragraph. A sentence has been added to highlight our planned future work. On lines 391-393.

P22 In 501: References were relatively old. Please update the literature review from 2018-2020.

Thank you for highlighting this. We have updated the references as suggested. We have also added the previously missing publication date (2019) for the Yoshida et al reference.

P31 Supp Table 1: What does IBS and IBD mean?

Thank you for highlighting this. We have replaced the abbreviations with the full terms in all tables where they had previously been abbreviated.

P40 Supp Table 7 legend: No * has been seen in Suppl Table 7

Thank you for highlighting this. The * description from Supplementary Table 7 Figure legend has been removed. Further edits have been made to the presentation of the table – placing brackets around 95% confidence intervals and labelling HRs with a $p > 0.01$ with + instead of giving the specific p-values. We have also made some formatting changes (mainly the addition of brackets for 95% CI) to some of the other tables for consistency and ease of reading. [This also links to the last comment from reviewer 2].

Reviewer 2 comments

Is there any synergistic effect with RA or is it only additive? Please present the results so that this question is also addressed; e.g. in tables or in suppl fig,

We have presented results from interaction analysis to examine whether there was a multiplicative or synergistic effect between individual LTCs and RA in their association with mortality or MACE. We have edited the wording in our findings (lines 292-294) to make this distinction between additive and synergistic effects clear: Interaction terms for RA and osteoporosis showed no significant interaction with all-cause mortality ($p=0.10$), suggesting an additive effect only, but displayed a significant interaction with MACE ($p < 0.01$), suggesting a multiplicative or synergistic effect in the association with MACE. This was also highlighted in the discussion; edits have now been made to lines 317 and 320 to make clear that our use of the term multiplicative is the same as synergistic.

VERSION 2 – REVIEW

REVIEWER	Wenhui Zhang Emory University, US
REVIEW RETURNED	14-Sep-2020
GENERAL COMMENTS	Thanks for asking me to review the manuscript again. This manuscript has been improved. One sentence was noticed that needs revision: "To investigate sensitivity of self-report by participants with RA, was examined the proportion of people with any primary care RA Read code, any secondary care RA hospitalisation code,..."